# Preparation and Pore Structure of Energy-Storage Phosphorus Building Gypsum

**DOI:** 10.3390/ma15196997

**Published:** 2022-10-09

**Authors:** Shixiong Liao, Kun Ma, Zhiman Zhao, Lei Wu, Zhuo Liu, Sicheng Quan

**Affiliations:** 1Faculty of Civil Engineering and Mechanics, Kunming University of Science and Technology, Kunming 650500, China; 2Yunnan Ningchuang Environmental Protection Technology Co., Ltd., Anning 650300, China; 3Faculty of Architecture and Engineering, Yunnan University of Economics and Management, Anning 650300, China

**Keywords:** energy storage, paraffin, phosphorous building gypsum, response surface, pore structure

## Abstract

In this study, the pore structure of a hardened phosphorous building gypsum body was optimised by blending an air-entraining agent with the appropriate water–paste ratio. The response surface test was designed according to the test results of the hardened phosphorous building gypsum body treated with an air-entraining agent and an appropriate water–paste ratio. Moreover, the optimal process parameters were selected to prepare a porous phosphorous building gypsum skeleton, which was used as a paraffin carrier to prepare energy-storage phosphorous building gypsum. The results indicate that if the ratio of the air-entraining agent to the water–paste ratio is reasonable, the hardened body of phosphorous building gypsum can form a better pore structure. With the influx of paraffin, its accumulated pore volume and specific surface area decrease, and the pore size distribution is uniform. The paraffin completely occupies the pores, causing the compressive strength of energy-storage phosphorous building gypsum to be better than that of similar gypsum energy-storing materials. The heat energy further captured by energy-storage phosphorous building gypsum in the endothermic and exothermic stages is 28.19 J/g and 28.64 J/g, respectively, which can be used to prepare energy-saving building materials.

## 1. Introduction

Energy consumption has increased with the rapid economic growth, and its main form is building energy consumption [1,2]. At present, heat- and energy-storage materials are widely used in energy-saving building materials to alleviate the problem of building energy consumption [3]. Phase-change materials can store and release a large amount of heat energy in the phase-change process [4,5], which can improve thermal comfort. With high heat-storage capacity and good thermal stability, they are considered to be among the most promising heat-storage and energy-storage materials [6,7,8]. Among them, paraffin phase-change materials have been widely used because of the high latent heat of the phase change, the absence of undercooling and chromatography, and being non-toxic, non-corrosive, and inexpensive, among other advantages. A new type of energy-storage building material that not only retains the advantages of the original material, but also inherits the properties of the phase-change material, can be obtained by combining the phase-change material with the traditional building material in a certain process. Phosphorus building gypsum has broad prospects for use as an energy-storage building material. Domestic and foreign scholars have studied many new energy-storage gypsum materials.

The current growth rate of phosphorus gypsum is estimated to be 200 million tonnes per year, whereas the effective utilisation rate is only 10–15%, according to the relevant statistical prediction [9]. The storage capacity is still increasing, causing considerable environmental pollution [10,11]. Currently, one of the most effective methods is to prepare phosphorous building gypsum for plasterboards, road building, cement retarders, and soil modifiers [12,13,14] via water washing, citric acid treatment, lime neutralisation, and high-temperature dehydration of phosphorus gypsum. However, the added value of phosphorous building gypsum products is relatively low owing to their poor mechanical properties and quality. Therefore, some researchers [15,16,17,18,19] have tried to use phosphorous building gypsum as an energy-storage building material to better improve its added value. Hua F et al. [15] used Ca-P/EG as the carrier of a paraffin phase-change material and gypsum composite to prepare a phase-change gypsum board. At 20% Ca-P/EG content, the gypsum board showed good thermal stability after 400 cycles of melting and freezing, but its compressive strength decreased by 44.87%. Liu Z.Y. et al. [16] used red mud as a paraffin phase-change material carrier and compounded gypsum to prepare energy-storage gypsum products. The thermal performance of the energy-storage gypsum significantly improved at 30% content of the paraffin/red mud composite phase-change material. The compressive strength dropped by 66.67%. Li Y. et al. [17] synthesised a composite phase-change material (CPCM) with steel fibre as the paraffin carrier and made a phase-change energy-storage gypsum board with gypsum as the cementing material. The energy-storage effect was the best when the CPCM content was 33%. The compressive strength, however, decreased by 80%. Chen X. et al. [18] used modified bentonite as a binary mixed-acid phase-change material of lauric and capric acids to prepare a stable CPCM and compounded it with gypsum to prepare a gypsum-based phase-change energy-storage material. The temperature range of the gypsum-based phase-change energy-storage material mixed with 50% CPCM could reach 7.4 °C and 12.4 °C in the process of heat storage and release, respectively. Its compressive strength was 1.6 MPa, which was 89.81% lower than that of the blank group. Meanwhile, Zhao L. et al. [19] prepared a phase-change microcapsule with paraffin as the core material and gelatinised flour as the wall material through the adsorption and deposition methods. The phase-change temperature, latent heat value, and coating rate of the phase-change microcapsule were 22.2 °C, 110.5 J/g and 66.7%, respectively. Adding microcapsules in the way of the wet material was beneficial in reducing the mechanical strength loss of the phase-change gypsum test block; however, the compressive strength still decreased by 45.7%.

In the abovementioned studies [15,16,17,18,19], the poor mechanical properties of energy-storage gypsum were mainly caused by the low strength of the porous phase-change material and the weak combination of the energy-storage aggregate encapsulating the material and gypsum. The leakage of the phase-change material in the energy-storage aggregate interfered with the hydration of the gypsum [20]. Currently, methods for improvement are to add nanomaterials to the energy-storage materials or to store the phase-change materials in the form of microcapsules [21,22]. However, these two methods are expensive and complex. Therefore, researchers have proposed a shape-setting phase-change material method [23,24,25] to store the phase-change materials with their own structure. This method is a simple and cost-saving process that can improve strength. However, a gypsum hardened body has a poor pore structure [26], less adsorbed phase-change material, and low potential. Therefore, it is important to develop a gypsum skeleton with a good pore structure to set the shape of the phase-change material and improve the mechanical properties of energy-storage gypsum.

The response surface methodology (RSM) is a combination of mathematics and statistics that can design experiments, construct models, analyse the influence of independent variables, and optimise experimental process parameters [27,28]. It can be used for evaluating the importance of some influential parameters in complex interactions and obtaining the optimal parameters more effectively and economically [29]. The RSM is widely used for optimising experimental parameters in cementitious materials [30,31,32,33]. One of its disadvantages, however, is the need to verify the extent of influence of a single factor on the material to design a better model for predicting the performance. In this study, the air-entraining agent and water–paste ratio were used to explore the paraffin adsorption rate and compressive strength of a hardened phosphorous building gypsum body. Because of the influence of the environment and parameters, finding the appropriate ratio is difficult and the experimental process is long, which is not conducive to the experimental design. Therefore, in this study, after determining the influence of a single factor on the performance of the hardened phosphorous building gypsum body, the response method was used to design the experiment, and the optimal ratio of the gypsum skeleton was studied.

In summary, this study puts forward a new idea of using the water–paste ratio and an air-entraining agent to optimise the pore structure of the hardened phosphorus building gypsum (PBG) body and use it to store phase-change materials. Through the central composite response surface model design and significance evaluation, the optimal process conditions are selected to prepare the porous phosphorus building gypsum (PPBG) used as the paraffin carrier to prepare the energy-storage phosphorus building gypsum (ESPBG) and further explore the pore structure of the porous phosphorus building gypsum to obtain a new energy-storage gypsum with the basic strength requirements. The microstructure, pore size distribution, cumulative pore volume, pore type, and pore structure of the porous phosphorous building gypsum and energy-storage phosphorous building gypsum are studied through scanning electron microscopy (SEM) and the Brunauer–Emmett–Teller (BET) method. The compatibility, chemical characterisation, and thermal properties of the stored phosphorous building gypsum are investigated through Fourier-transform infrared (FTIR) spectroscopy, X-ray diffraction (XRD), and differential scanning calorimetry (DSC) to ensure the feasibility of the optimised pore structure’s application in the PBG.

## 2. Materials and Methods

### 2.1. Raw Materials

The phosphorous building gypsum produced by Yunnan Yantianhua Co., Ltd. (Kunming, China). was used in this study. The phosphorous building gypsum was made from phosphorus gypsum by water washing, citric acid treatment, lime neutralisation, and dehydration at 145 °C for 6 h. The chemical composition of the phosphorous building gypsum was characterised by X-ray fluorescence spectrometry. Table 1 presents the test results.

The cellulose ether (foam stabiliser) was a 200,000-viscosity cellulose ether produced by Zhongshan Huizhong Chemical Technology Co., Ltd. (Zhongshan, China). The paraffin used was liquid paraffin manufactured by Dongguan Shengbang Plastic Co., Ltd. (Dongguan, China); the main elements in the paraffin were C, H, and O, and the main minerals were (CH_2_)x, C_46_H_94_, and C_16_H_34_O [16]. The air-entraining agent was rosin concrete air-entraining agent (a pore-forming agent) produced by Shenyang Shengxinyuan Building Materials Co., Ltd., and the main elements in it were C, H, and O. Ordinary tap water was used as the mixing water.

### 2.2. Preparation and Characterisation of Energy-Storage Phosphorous Building Gypsum

#### 2.2.1. Single-Factor Experiment

First, phosphorous building gypsum slurry was prepared according to the experimental mix proportions presented in Table 2.

Following the specifications stated in the ‘Determination of Mechanical Properties of Building Gypsum’ (GB/T17669.3-1999), part of the phosphorous building gypsum slurry was poured into a 40 × 40 × 160 mm triple die to test its compressive strength. The other part was poured into a 20 × 20 × 20 mm six-joint cement paste die to test the paraffin absorption rate. The experimental samples were removed after 24 h and cured to a constant weight at a constant temperature of 50 °C.

#### 2.2.2. Compressive Strength Test

Following the specifications stated in the ‘Determination of Mechanical Properties of Building Gypsum’ (GB/T17669.3-1999), 40 × 40 × 160 mm test blocks with an optimal ratio in the single-factor experiment and the central composite RSM were tested using a cement compression machine. The loading rate was 0.8 kN/s.

#### 2.2.3. Adsorption Rate Test

The adsorption process was adopted to determine the paraffin adsorption rate, as shown in Figure 1.

The prepared 20 mm × 20 mm × 20 mm phosphorous building gypsum test blocks were introduced into the dryer and then vacuum-dried for 20 min with a closed phase-change paraffin valve. The paraffin valve was opened in the process of dipping the phase-change paraffin. The paraffin was then finally absorbed into the dryer with the test block under a negative pressure (−0.06 MPa) in a water bath at a constant temperature (50 °C). The previous valve was kept closed. When the dryer was dipped for 30 min without obvious bubbles, the specimens were taken out and cooled to a normal temperature for moulding, consequently obtaining the ESPBG. The specimens were then baked in an oven at 50 °C for 4 h to release the excess paraffin from the test block pores and prevent leakage and damage to strength during use [22]. Equation (1) was used to calculate the paraffin absorption rate of the phosphorous building gypsum test block [34]:(1)$$W_{absorption}=(W_{1}-W_{2})/W_{2}\times 100\%$$
where, $W_{1}$ is the mass (g) in the saturated state of the material absorption and $W_{2}$ is the mass of the material in the dry state (g).

#### 2.2.4. Microstructure Test

The microstructure of the test block containing PBG, PPBG, and ESPBG was analysed using a ZEISS Gemini 300 SEM in Japan. Prior to observation, the samples were dried and gold-plated in an ion sputter coater for 2–5 min, with an accelerating voltage of 3–20 kV.

#### 2.2.5. Thermal Performance Test

The phase transition temperature and thermal energy of the paraffin and ESPBG blocks were measured via differential scanning calorimetry (DSC), where the block size should not be >3 mm in diameter and >2 mm in height. Paraffin wax needed to be frozen into solid paraffin wax before testing. In the test process, a standard crucible (pure aluminium) and a nitrogen atmosphere were used. The isothermal curves of the endothermic stage at 0–55 °C and the exothermic stage at 55–0 °C were maintained, and the heating and cooling rates were constant and decreased by 2 °C/min. The test time was 2 h.

#### 2.2.6. Pore Structure Test

The pore structure data tests were conducted with a fully automatic BET specific surface and porosity analyser (model: MAC ASAP2460). The PBG, PPBG, and ESPBG test blocks were analysed. The test conditions were the data of all pores (i.e., specific surface area plus pore size distribution, including mesopores and micropores) and the adsorption–desorption curves of N_2_ adsorbed at a degassing temperature of 120 °C for 4 h.

#### 2.2.7. Composition Test

The phase composition of the PBG, PPBG, and ESPBG was analysed using an X-ray diffractometer. The samples were all in powder form, and the test was in the range of 10–100° with a scanning speed of 10°/min.

#### 2.2.8. Compatibility Test

The absorbance of the paraffin and the PBG, ESPBG, ESPBG powders after 100 hot and cold cycles was measured in the wavenumber range of 400–4000 using Bruker’s MPA and Tensor 27.

#### 2.2.9. Thermal Cycling Test

The high–low-temperature alternating box (HD-E702-100) of Dongguan Haida Instrument Co., Ltd. (Dongguan, China), was used to test the stability of the hot cycle of ESPBG. The prepared high–low cycle samples were placed on a plate and then moved to the low-temperature alternating box, at temperatures of 20–65 °C. The heating temperature was stabilised and maintained for 20 min, followed by 100 cycles, and then removed. The functional groups were analysed via Fourier-transform infrared spectroscopy.

## 3. Results and Discussion

### 3.1. Single-Factor Experiment Result Analysis

Figure 2a shows the relationship of the water–paste ratio, compressive strength, and adsorption rate of the phosphorous building gypsum in the single-factor experiment.

In the 0.5–0.60 range, the water–paste ratio was inversely proportional to the compressive strength and directly proportional to the absorption rate. The corresponding compressive strength was 14.34–10.61 MPa, and the corresponding absorption rate was 10.29–13.22%. Figure 2b depicts the relationship of the air-entraining agent, compressive strength, and adsorption rate of the phosphorous gypsum. In the 0–1% range, the content of the air-entraining agent was negatively correlated with the absolute dry compressive strength and positively correlated with the adsorption rate. The corresponding compressive strength was 10.61–4.37 MPa, and the corresponding adsorption rate was 13.22–27.62%. At the same time, within the range of satisfying the functional relationship, the 2 h strength of the two single factors was within the requirements of the building gypsum standard (GBT 9776-2008); hence, these factors can be used to prepare gypsum products.

### 3.2. Central Composite Response Surface Model Design and Significance Evaluation

The experimental factors of the central composite response surface were designed according to the single-factor experiment results and Equation (2). The horizontal level was set as 0. The two sides of the horizontal level were set as −1 and 1, with the compressive strength and the adsorption rate as the response values. Table 3 depicts the level table. Table 4 shows the experimental results of the central composite response surface.

This experiment had 13 experimental points, 4 of which were zero points, while the other 9 were analysis factors. Response value *Y*_1_ (Equation (3)) denotes the absolute dry compressive strength, while response value *Y*_2_ (Equation (4)) represents the adsorption rate. The compressive strength and the adsorption rate in Table 3 were fitted using Design-Expert V8.0.5 software. The following regression equation was obtained:(2)$$Y=\beta_{0}+\sum_{i=1}^{k}\beta_{i}x_{i}+\sum_{i=1}^{k}\beta_{ii}x_{i}+\sum_{\begin{array}{l} i=1\\ i<j \end{array}}^{k}\beta_{ij}x_{i}x_{j}$$
where β is the unknown coefficient; k is the number of design variables; Y is the predicted response value; and $\beta_{0}$, $\beta_{i}$, and $\beta_{ii}$ are the migration term, linear migration, and second-order migration coefficients, respectively; $\beta_{ij}$ is the interaction coefficient.
(3)$$Y_{1}=7.70-2.27X_{1}-3.05X_{2}+0.13X_{1}X_{2}+0.43X_{1}{}^{2}+1.45X_{2}{}^{2}$$
(4)$$Y_{1}=20.19+2.87X_{1}+3.40X_{2}+0.74X_{1}X_{2}+2.31X_{1}{}^{2}-6.49X_{2}{}^{2}$$

Table 5 and Table 6 present the results of the variance analysis of the quadratic regression model obtained using Design-Expert V8.0.5.

According to the analysis of variance in Table 5 and Table 6, F_1_ = 17.70 and P_1_ = 0.0008 < 0.01 in the quadratic regression model of the compressive strength, while F_2_ = 17.20 and P_2_ = 0.0008 < 0.01 in the quadratic regression model of the adsorption rate. These results show that the relationship between Y_1_ and Y_2_ and the equations in the quadratic regression model are extremely significant in accordance with the statistical law. P_1_ and P_2_ in the misfitting items were >0.05; hence, the model had a certain reliability. In the quadratic regression model of Y_1_, the order of the absolute value of the first-term coefficient was X_2_ > X_1_, showing the degree of influence that the two factors have on the absolute dry compressive strength, i.e., air-entraining agent > water–paste ratio. The order of the absolute value of the first-term coefficient in the quadratic regression model of Y_2_ was X_2_ > X_1_, showing the degree of influence that the two factors have on the adsorption rate, i.e., air-entraining agent > water–paste ratio.

### 3.3. Analysis of the Three-Dimensional Surface and Contour in the Response Surface

Figure 3 and Figure 4 depict the response surface curves and contours, respectively, of the influence of the water–gypsum ratio and the air-entraining agent interaction on the compressive strength and absorption rate of the phosphorous gypsum according to the response surface analysis method [35].

Figure 3a and Figure 4a illustrate that under the condition of an unchanged water–paste ratio, the compressive strength gradually decreased with the increase in the air-entraining agent content, and then tended to stabilise. The air-entraining agent content remained unchanged under these circumstances. With the increase in the water–paste ratio, the compressive strength gradually decreased and then stabilised. The figures also show the two influences on the compressive strength trend, i.e., air-entraining agent > water–paste ratio, in line with the law of regression equation Y_1_. Figure 3b and Figure 4b show that under the interaction, the influence of the two factors on the adsorption rate presents a convex curved surface. Under the condition of a constant water–paste ratio, the adsorption rate presented a trend of first rising and then falling with the increase in the air-entraining agent content. With the increase in the water–paste ratio, the adsorption rate presented a gradual upward trend under the condition of a constant air-entraining agent dosage. The results showed that the influence trend of the air-entraining agent > water–paste ratio on the adsorption rate was in accordance with the law of regression equation, Y_2_.

Figure 5a,b show the evaluation lines of the predicted values of the compressive strength and the adsorption rate.

The closer the point was to the diagonal line, the closer the predicted value was to the experimental value. The two groups of data were close to one another, verifying the reliability of the model. In the design of the central composite response surface model using Design-Expert V8.0.5 software, the importance degree of the adsorption rate was designed as +++, whilst that of the absolute dry compressive strength was designed as ++. Figure 3 and Figure 4 show the following optimal process conditions: water–paste ratio = 0.6; air-entraining agent content = 0.66%; predicted adsorption rate = 26.03%; and predicted compressive strength = 4.38 MPa. The experimental value of the adsorption rate was 26.06%, while that of the compressive strength was 4.40 MPa. The experimental values were in good agreement with the predicted values, proving that the model has a certain reliability and practicability.

### 3.4. Compressive Strength of the PBG, PPBG, and ESPBG

The hardened phosphorous building gypsum body is expressed as the PBG. The porous phosphorus building gypsum prepared using the optimal parameters determined via the central composite response surface model design method is expressed as the PPBG. This absorbs the energy-storage phosphorus building gypsum prepared with the paraffin, expressed as the ESPBG. Figure 6 depicts the compressive strength comparison of the PBG, PPBG, and ESPBG.

Under the same mixture ratio, the compressive strength of the PPBG decreased by 59.50% compared with that of the PBG due to the pore structure optimisation of the PBG by the water–paste ratio and the air-entraining agent, which resulted in an increase in porosity and a decrease in strength. Compared with those of the PPBG and the PBG, the compressive strength of the ESPBG increased by 75.65% and decreased by 28.85%, respectively, due to the phase transition temperature of the paraffin wax having a small change range. The PPBG’s porosity was also reduced after filling into the pore structure at a normal temperature. It existed in solid form, as can also be seen from the microstructure in Figure 6. Compared with the PBG, part of the strength decreased after the paraffin filling. This was caused by some of the closed pores in the PBG, which could not be filled. However, compared with the existing energy-storage gypsum materials, the ESPBG has better compressive strength and a better decline rate. Figure 7a,b present the comparison results, showing that the ESPBG has a certain applicability.

### 3.5. Microstructure Analysis

Figure 8 depicts the microscopic morphology of the PBG, PPBG, and ESPBG. Figure 8a presents an SEM image of the PBG, illustrating that the PBG has a flat outline with no obvious pores.

When enlarged, the figure showed a small number of pores. The SEM images of the PPBG at different scales (Figure 8b,c) show that the PPBG has an obvious pore structure and some cylindrical pores, which can only be seen when the figure is enlarged. Comparing Figure 8a with Figure 8b,c, the pores were found to increase. This indicates that the PBG hole pattern changed and the pore structure was good after the optimisation of the water–paste ratio and air-entraining agent. In other words, the PPBG is an excellent porous skeleton, which is greatly significant for phase-change material storage. Figure 8d displays an SEM image of the ESPBG. Compared with Figure 8c, we found almost no visible pores after the paraffin impregnation. The swirling macropores were also filled with paraffin, indicating that paraffin was fully immersed in the pores of the PPBG. However, some pores were still insufficiently filled. One of the main reasons for this was that the PPBG contained some closed pores and the paraffin could not be filled.

### 3.6. Thermal Performance Analysis of Paraffin and ESPBG

Figure 9a,b show the DSC curves of the paraffin wax and the ESPBG, respectively.

The melting temperature of the paraffin wax was 15.6 °C, which increased to 18.50 °C in the ESPBG, showing an increase of 2.90 °C. For the freezing temperature, the paraffin crystallisation temperature was 29.264 °C, which increased to 27.14 °C in the ESPBG, showing a reduction of 2.124 °C. Previously, some researchers have linked the change in the phase transition temperature (T_melt_/T_freeze_) of energy-storage materials with the combination of the carrier of the energy storage materials and the paraffin [36,37]. If the combination between the first two is cohesive, the result will be an increase in the phase transition temperature, and vice versa. Compared with paraffin, the melting and freezing temperatures in the ESPBG both increased, indicating that the combination of the PPBG and paraffin has a certain mutual attraction. Figure 9a,b also show that the DSC curve trends of the paraffin and the ESPBG were similar, indicating that T_melt_ and T_freeze_ were the inherent characteristics of paraffin phase-change materials. The heat energy of the paraffin was 108.17 J/g in the endothermic stage and 103.6 J/g in the exothermic stage. The heat energy of the ESPBG in the endothermic and exothermic stages was 28.19 J/g and 28.64 J/g, respectively, reflecting the paraffin heat energy retained in the ESPBG (26.06%). The energy-storage effect effectively reached the thermal performance of the energy-storage aggregate and the gypsum board in the literature [15,17,38], and can be used for building energy-saving materials, subsequent PPBG structure, energy storage aggregates, and gypsum board.

### 3.7. Pore Structure Analysis

#### 3.7.1. Adsorption–Desorption Isotherm Analysis of the PBG, PPBG, and ESPBG

Figure 10a shows the N_2_ adsorption–desorption isotherms of the PBG, PPBG, and ESPBG, reflecting the specific surface area and the pore structure of the materials [39]. At P/P_0_ (relative pressure) of 0–0.2, the adsorption capacities of the PBG, PPBG, and ESPBG were stable with the increase in pressure, indicating that the microporous structure was rare or absent.

For 0.2 < P/P_0_ < 0.9, the adsorption isotherm curve of the PBG slowly changed and did not coincide with the desorption curve. The main reason for this result may be the condensation of some capillary pores. Figure 10b depicts the pore structure mainly comprising slits pores (i.e., open on both ends). The slits pores were caused by the irregular overlap between the flake and the rod crystals in the PBG. Two types of slits pores exist: open and closed. The adsorption isotherm curve of the PPBG quickly changed and did not coincide with the desorption curve. This type of curve reflected a typical cylindrical pore structure (i.e., opening and closing), as shown in Figure 10c. The main reason for this result is that condensation and decondensation may need to be completed within the effective hole radius; hence, the trend of the adsorption and desorption curves is inconsistent. The adsorption isotherm curve of the ESPBG coincided with the desorption curve, indicating that the pores in the PPBG were filled with the absorbed paraffin. When P/P_0_ > 0.9, the adsorption curve of the phosphorous building gypsum increased in a small range and the slope was smaller, showing that the phosphorous building gypsum contained mesopores. The adsorption curve of the PPBG increased in a small range with a large inclination, indicating that it contained mesoporous and macroporous structures. Meanwhile, the adsorption curve of the ESPBG was almost unchanged, showing that the pore structure was difficult to form. Figure 10 also shows the specific surface areas of the PBG, PPBG, and ESPBG as 3.25 m^2^/g, 14.64 m^2^/g, and 0.22 m^2^/g, respectively; the total pore volumes were 0.031 cm^3^/g, 0.068 cm^3^/g, and 0.002 cm^3^/g, respectively. The micropore volume in the PBG was 0.000213 cm^3^/g, accounting for approximately 0.67% of the total volume.

#### 3.7.2. Pore Size Distribution Analysis of the PBG, PPBG, and ESPBG

Figure 11a,b depict the pore size distribution curves of the PBG, PPBG, and ESPBG.

Figure 11a,b show that the pore size distributions of the PBG, PPBG, and ESPBG were 1.74–27.71 nm (1 Å = 0.1 nm), 2.19–100 nm, and 4.35–33.2 nm, respectively. The results in Figure 11a,b show that the cumulative pore volumes of the PBG, PPBG, and ESPBG in the pore diameter range of 0–1000 nm were between 0 and 0.022 cm^3^/g, 0 and 0.042 cm^3^/g, and 0 and 0.0019 cm^3^/g, respectively. The results in Figure 12a,b illustrate that the PBG was mainly mesoporous, with fewer micropores and macropores.

The cumulative pore volume and the amount of paraffin that could be stored were small. The PPBG results showed that it was mainly composed of macropores and contained fewer mesopores. In their research, Yin et al. [40] pointed out that the pore size corresponding to the peak in the pore size distribution curve was more likely to appear. Moreover, the PPBG would have more 2.5 nm mesopores and 53.81 nm and 90.45 nm macropores in the pore size range of 2.19–100 nm under a better cumulative porosity. The analysis data of the PBG, PPBG, and ESPBG illustrated that the PPBG has an excellent pore structure. With the addition of paraffin, the total pore volume of the ESPBG, as well as the pore size distribution and the specific surface area, gradually decreased, indicating that paraffin flowed into the PPBG pores.

Additionally, the pore type, pore size distribution, and cumulative pore volume of the hardened gypsum body can affect the relationship between the microstructure and the mechanical properties. The slits pores in PBG are dense, and the corresponding hole distribution interval and pore volume are small, according to Figure 8a, Figure 11a, and Figure 12a. These results sharply contrast with the optimal strength of PBG given in Figure 6. An increase in cylindrical pores in PPBG improved the pore size distribution and pore volume of PBG, making it possible to store more paraffin wax and reduce its strength. However, the pore structure was filled with the influx of 26% paraffin into the open cylindrical pores of the PPBG. Almost no pore structure was observed (Figure 8d). The appropriate paraffin storage amount in ESPBG overcame the poor mechanical properties of PPBG and, eventually, endowed it with better mechanical properties.

### 3.8. XRD Analysis of the PBG, Paraffin, and ESPBG

Figure 13 illustrates the XRD patterns of the PBG, PPBG, and ESPBG. The main components of PBG are CaSO_4_·2H_2_O and a few quartz impurities. An XRD standard library search (number of JCPDs: 41-00244) showed that the PBG had strong diffraction peaks near 2θ = 8.6°, 19.8°, 30.14°, and 22.5°; the main elements in the paraffin are C, H, and O, and the main minerals are (CH_2_)x, C_46_H_94_, and C_16_H_34_O [16]; the rosin concrete air-entraining agent’s (i.e., pore-forming agent) main elements are C, H, and O. After the rosin-based concrete air-entraining agent was added to the PBG slurry, stable microbubbles were introduced into the PBG slurry through physical action to form a PPBG pore structure, and then the ESPBG was prepared by adsorbing paraffin. The main components of PPBG are CaSO_4_·2H_2_O and a few quartz impurities; PPBG had strong diffraction peaks near 2θ = 8.55°, 19.6°, 31.02°, and 23.55. The main components of ESPBG are CaSO_4_·2H_2_O and a few quartz impurities; ESPBG had strong diffraction peaks near 2θ = 8.75°, 19.8°, 31.61°, and 24.01°.

The ESPBG contained all of the peaks of PBG and PPBG, but the peak intensities were relatively lower in comparison with those of PBG and PPBG. The results showed that the addition of a small amount of air-entraining agent in PPBG only formed bubbles, making the PBG form a good pore structure, and the peak value of the main components changed little. However, due to the adsorption of 26.06% paraffin in ESPBG, the peak value of the material combined with paraffin was reduced [16]. Therefore, the peak intensities were relatively lower in comparison with those of PBG and PPBG, but no new peaks were generated in ESPBG, meeting the energy-storage material requirements.

### 3.9. FTIR Analysis of PBG, PPBG, and ESPBG

FTIR spectroscopy was used to analyse the PBG, paraffin, ESPBG, and ESPBG after thermal cycling. Figure 14 shows the results. The spectra clearly show that the infrared peaks of the ESPBG and the ESPBG after thermal cycling retain some characteristic absorption peaks of PBG and paraffin, and the peak shapes are broadly similar. The characteristic absorption peaks and peak trends of the PBG, paraffin, ESPBG, and ESPBG after thermal cycling can be seen in Table 7. The peak of the paraffin is smooth, and the characteristic peaks of paraffin are shown in Table 7. These peaks represent O–H bonding (2953.55 cm^−1^ and 2911.87 cm^−1^) and C–O bonding (1458.11 cm^−1^ and 1326.50 cm^−1^).

It can be seen from Figure 14 and Table 7 that the PBG peaks are generally smoother and the peaks are sharp at a few wavenumbers. These peaks represent O–H stretching vibration (3546.27 cm^−1^ and 3403.62 cm^−1^), S–O bending vibration (1684.33 cm^−1^ and 1618.23 cm^−1^), O–H symmetrical vibration (1127.10 cm^−1^), and C–H stretching (659.24 cm^−1^ and 601.33 cm^−1^). As shown in Table 7, the characteristic peak positions and shapes of the ESPBG and the ESPBG after thermal cycling are effectively the same, with common characteristic peaks at 3547.84 cm^−1^, 2920.13 cm^−1^, 2849.80 cm^−1^, and 602.50 cm^−1^. The peak values of the ESPBG and the ESPBG after thermal cycling are also large and sharp near 2849.80 cm^−1^, 1685.94 cm^−1^, 1621.58 cm^−1^, and 670.91 cm^−1^. Among the smoother peaks of the ESPBG and the ESPBG after thermal cycling, there is a slight difference between 3547.84 cm^−1^, 3408.20 cm^−1^, 1459.22 cm^−1^, 1328.62 cm^−1^, and 1141.76 cm^−1^, showing that ESPBG has good thermal stability. The characteristic peaks of ESPBG are shown in Table 7. These peaks represent O–H stretching vibration (3547.84 cm^−1^–2849.80 cm^−1^), S–O bending vibration (1685.94 cm^−1^–1612.47 cm^−1^), C–O bonding (1460.20 cm^−1^–1327.42 cm^−1^), O–H symmetrical vibration (1141.76 cm^−1^–1139.26 cm^−1^), and C–H stretching (670.91 cm^−1^–602.50 cm^−1^).

The characteristic peaks of the raw materials (i.e., paraffin, PBG) were not changed in the ESPBG. A comparison of the characteristic spectra of the raw materials and ESPBG showed that there were no new peak shapes in the FTIR spectra of ESPBG. The peak locations were exactly superimposed on the PBG and paraffin [41]. Therefore, the phase-change energy-storage composites are stable.

## 4. Conclusions

In this study, the central composite response surface design was used to prepare the PPBG by selecting the optimal process parameters. The ESPBG was then prepared by using it as a paraffin carrier. The following conclusions were obtained according to the experimental results and basic analysis:The application of the response surface design of the central composite improved the analysis rate of the experimental data. The optimal process conditions of the central composite response surface design in this study were as follows: water–paste ratio = 0.6; air-entraining agent dosage = 0.66%; experimental value of the absorption rate = 26.06%; and experimental value of strength = 4.40 MPa. The material had a good pore structure (cylindrical pore), and there were many mesopores of 2.5 nm and macropores of 53.81 nm and 90.45 nm in the range of 2.19–100 nm. After absorbing paraffin, it was filled tightly, giving the energy-storage phosphorous building gypsum a good strength.The compatibility between ESPBG and paraffin was good, and the thermal energy in the endothermic stage and the exothermic stage was 28.19 J/g and 28.64 J/g, respectively, meeting the requirements of energy-saving building materials.

## 5. Potential Applications and Prospects of PPBG and ESPBG

Overall, the pore structure of the PPBG prepared in this work is worth popularising. The structure can be developed with light phosphorous building gypsum aggregate, light phosphorous building gypsum board, energy-storage phosphorous building gypsum aggregate, or energy-storage phosphorous building gypsum board. The research and development of these products will promote the utilisation of phosphorous building gypsum and realise the resource utilisation of phosphorous gypsum. The cost of gypsum products is very low, and the developed ESPBG is completely feasible among energy-saving materials. If utilised, it could alleviate the problem of the high energy consumption of building materials. The further problems of water resistance and the life cycle after cold and heat cycles in energy-storage gypsum materials need to be solved in future research. Then, in the future study of the pore structure of gypsum materials, mercury intrusion porosimetry (MIP) can be considered, which can enable the observation of more interesting pore structures.

## Figures and Tables

**Figure 1 materials-15-06997-f001:**
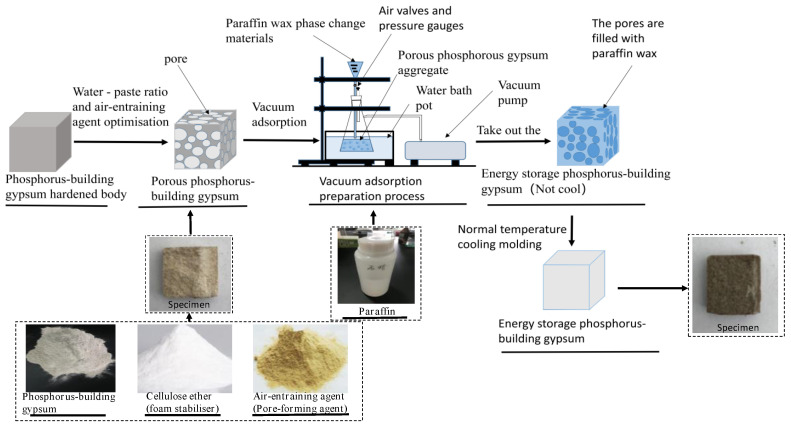
Schematic diagram of the vacuum adsorption process of the phosphorus gypsum test block.

**Figure 2 materials-15-06997-f002:**
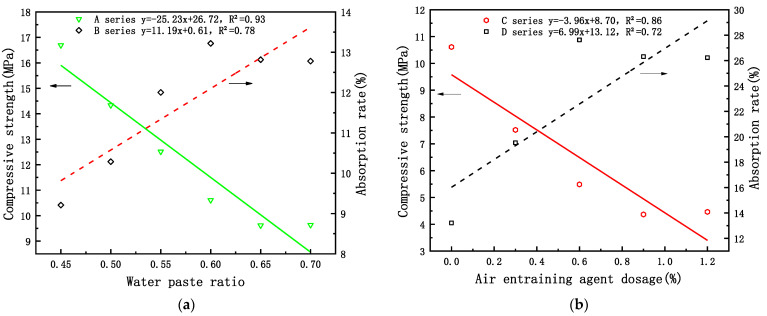
Relationship between the single factors and the absolute dry compressive strength and adsorption rate of phosphorous gypsum: (**a**) water–gypsum ratio; (**b**) air-entraining agent.

**Figure 3 materials-15-06997-f003:**
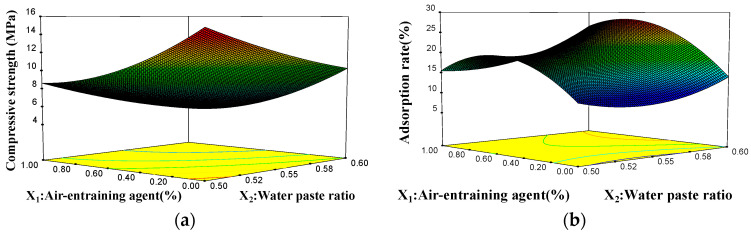
Response surface curves of the influence of the water–paste ratio and the air-entraining agent interaction on the compressive strength and adsorption rate of the phosphorous gypsum: (**a**) compressive strength and (**b**) adsorption rate response surface curves.

**Figure 4 materials-15-06997-f004:**
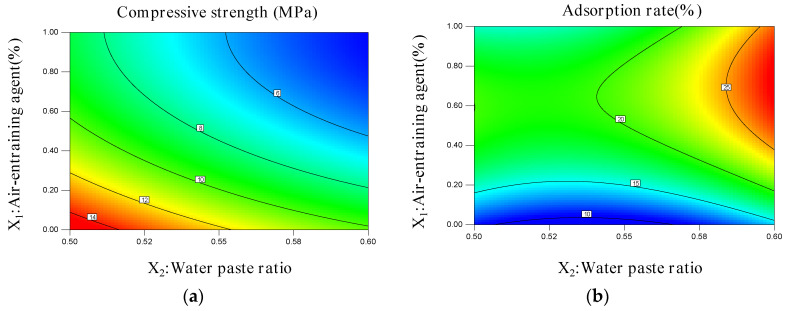
Response surface contours of the influence of the water–paste ratio and the air-entraining agent interaction on the (**a**) compressive strength and (**b**) absorption rate of the phosphorous gypsum.

**Figure 5 materials-15-06997-f005:**
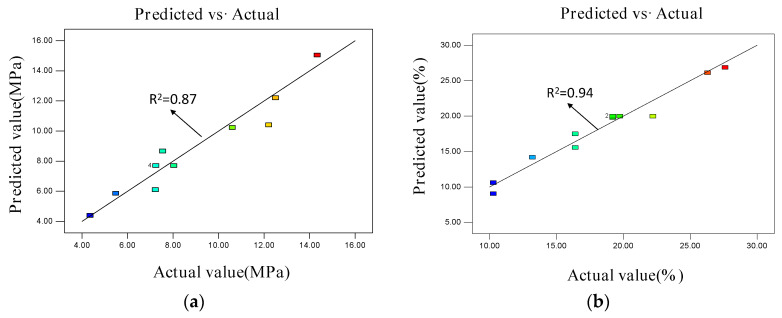
Predicted (**a**) compressive strength and (**b**) adsorption rate values of the phosphorous gypsum based on the water–paste ratio and the air-entraining agent interaction.

**Figure 6 materials-15-06997-f006:**
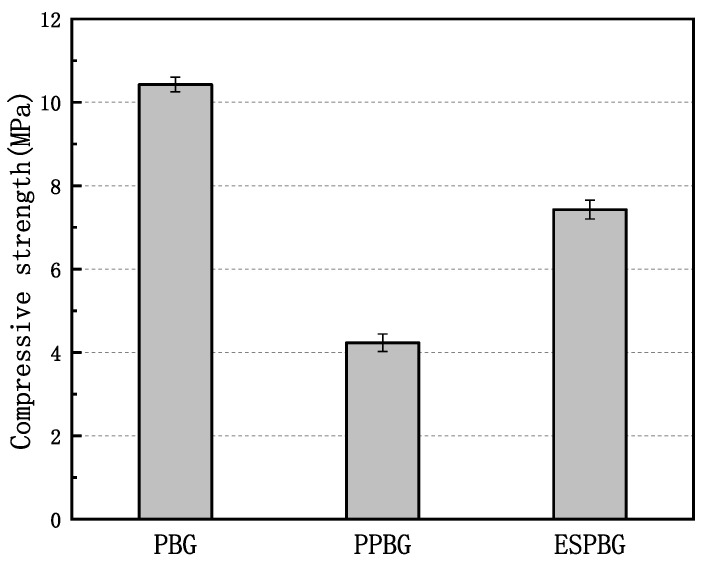
Compressive strengths of the PBG, PPBG, and ESPBG.

**Figure 7 materials-15-06997-f007:**
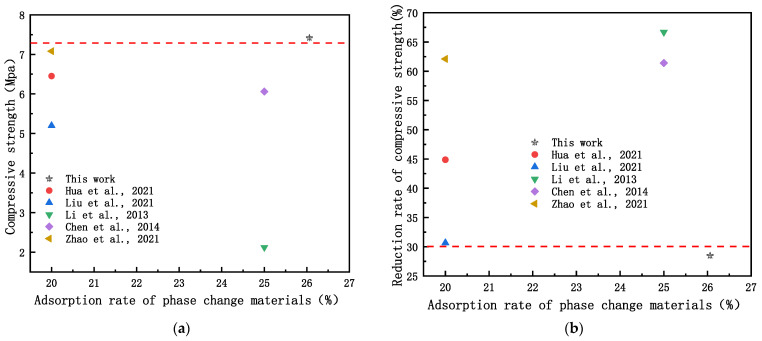
Comparison of existing studies: (**a**) compressive strength; (**b**) reduction rate of compressive strength [15,16,17,18,19].

**Figure 8 materials-15-06997-f008:**
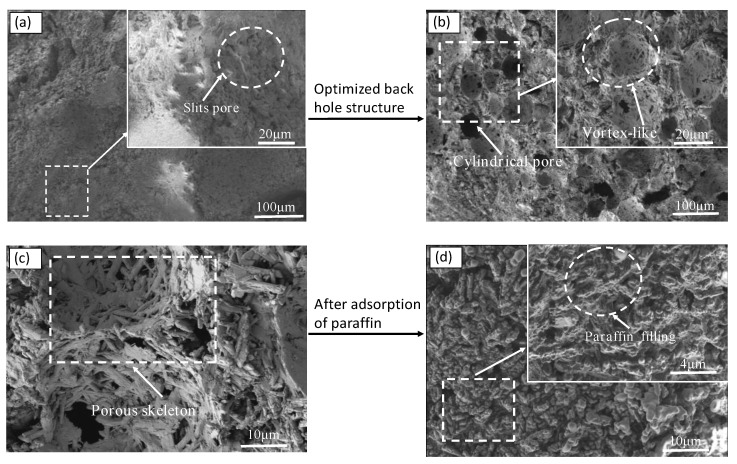
SEM images: (**a**) PBG; (**b**,**c**) PPBG; (**d**) ESPBG.

**Figure 9 materials-15-06997-f009:**
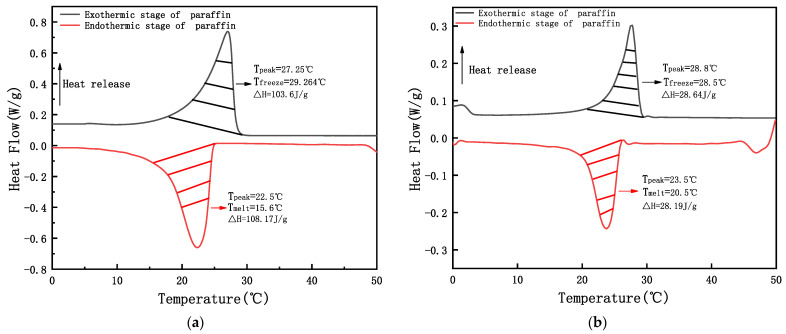
DSC curves of (**a**) paraffin and (**b**) ESPBG.

**Figure 10 materials-15-06997-f010:**
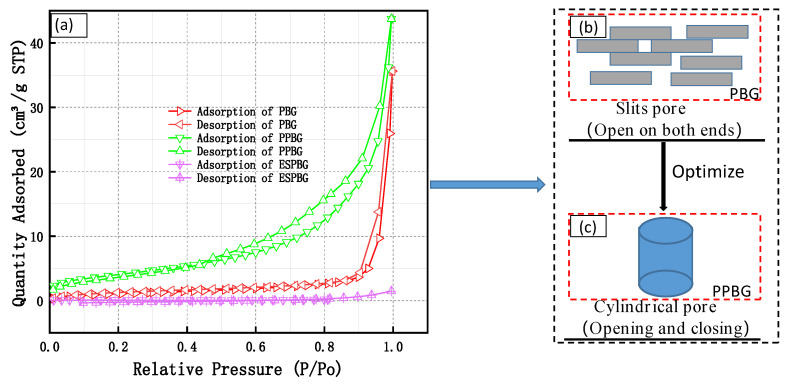
(**a**) Adsorption isotherm curves of the PBG, PPBG, and ESPBG; (**b**) PBG slits pore; (**c**) PPBG cylindrical pore.

**Figure 11 materials-15-06997-f011:**
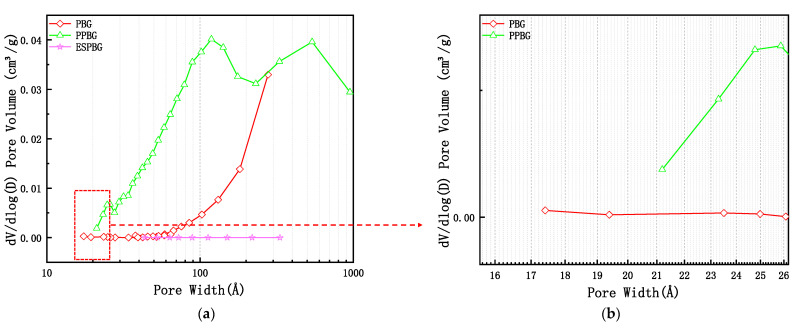
Pore size distribution of the PBG, PPBG, and ESPBG.

**Figure 12 materials-15-06997-f012:**
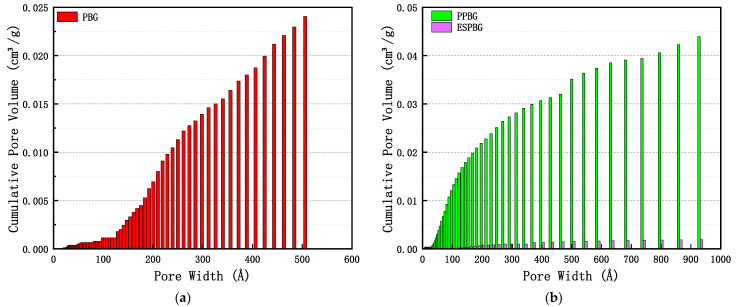
Cumulative pore volumes of the (**a**) PBG and the (**b**) PPBG and ESPBG.

**Figure 13 materials-15-06997-f013:**
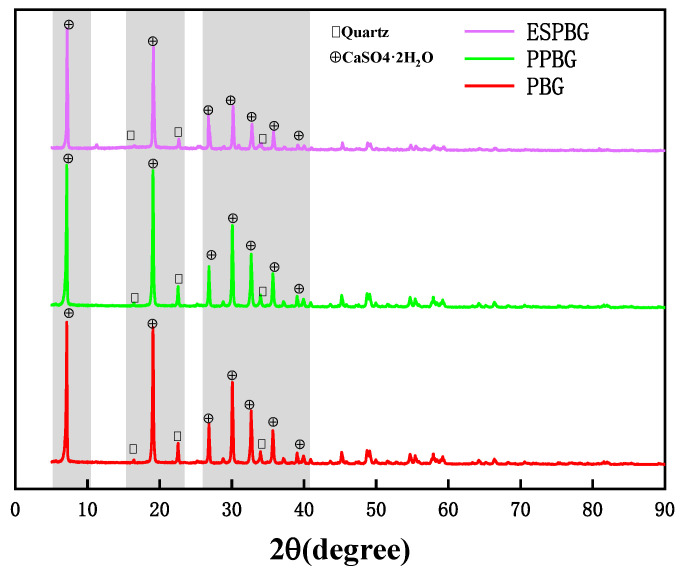
XRD patterns of the PBG, PPBG, and ESPBG.

**Figure 14 materials-15-06997-f014:**
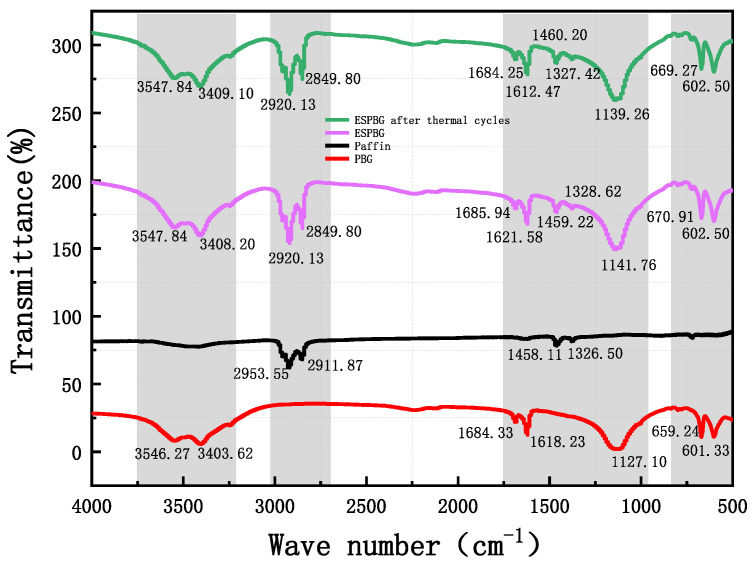
Infrared spectra of the PBG, paraffin, ESPBG, and ESPBG after thermal cycling.

**Table 1 materials-15-06997-t001:** Chemical composition of the phosphorous building gypsum.

Component	SiO_2_	P_2_O_5_	CaO	MgO	Al_2_O_3_	SO_3_	Fe_2_O_3_	K_2_O	F	Organic
Content (%)	14.25	0.61	31.05	0.14	0.12	43.1	0.16	0.26	0.23	0.22

**Table 2 materials-15-06997-t002:** Mixture ratio of phosphorus building gypsum.

Label	Phosphorous Building Gypsum (g)	Water–Paste Ratio	Air-Entraining Agent (g)	Cellulose Ether (g)
A_1_	1200	0.45	-	-
A_2_	1200	0.50	-	-
A_3_	1200	0.55	-	-
A_4_	1200	0.60	-	-
A_5_	1200	0.65	-	-
A_6_	1200	0.60	-	1.20
A_7_	1200	0.60	3.60	1.20
A_8_	1200	0.60	7.20	1.20
A_9_	1200	0.60	10.80	1.20
A_10_	1200	0.60	14.40	1.20

**Table 3 materials-15-06997-t003:** Central composite response surface factor-level table.

Factors	Symbolic Coding	Scope and Extent		
		−1	0	1
Water–paste ratio	X_1_	0.5	0.55	0.6
Air-entraining agent	X_2_	0	0.5	1

**Table 4 materials-15-06997-t004:** Experiment results of the central composite response surface.

Serial Number	Water–Paste Ratio	Air-Entraining Agent (%)	Compressive Strength (MPa)	Absorption Rate (%)
1	0.55	0.50	7.25	19.75
2	0.60	1.00	4.36	26.31
3	0.55	0.50	7.25	19.23
4	0.55	1.00	7.23	17.21
5	0.50	1.00	7.55	16.41
6	0.50	0.50	12.21	19.22
7	0.55	0.50	7.25	19.23
8	0.50	-	14.34	10.29
9	0.60	0.50	5.49	27.62
10	0.55	0.5	8.04	22.23
11	0.60	-	10.61	13.22
12	0.55	-	12.51	12.03
13	0.55	0.50	7.25	19.22

**Table 5 materials-15-06997-t005:** Variance analysis results of the quadratic regression model of Y_1_.

Source	Sum of Squares	Degrees of Freedom	Mean Squares	F_1_ Value	*p*_1_ Value > F1	Significant
Model	95.94	5	19.19	17.70	0.0008	Significant
X_1_	3.01	1	31.01	28.61	0.0011	-
X_2_	55.94	1	55.94	51.61	0.0002	-
X_1_X_2_	0.073	1	0.07	0.06	0.8028	-
X_1_^2^	0.51	1	0.51	0.47	0.5149	-
X_2_^2^	5.80	1	5.80	5.36	0.0539	-
Residual	7.59	7	1.08	-	-	-
Lack of fit	7.09	3	2.36	18.93	0.0079	Significant
Pure error	0.50	4	0.12	-	-	-
Cor total	103.52	12	-	-	-	-

**Table 6 materials-15-06997-t006:** Variance analysis results of the quadratic regression model of Y_2_.

Source	Sum of Squares	Degrees of Freedom	Mean Squares	F_2_ Value	*p*_2_ Value > F_2_	Significant
Model	237.37	5	47.47	17.20	0.0008	Significant
X_1_	49.48	1	49.48	17.92	0.0039	-
X_2_	69.29	1	69.29	25.10	0.0015	-
X_1_X_2_	2.21	1	2.21	0.80	0.4011	-
X_1_^2^	14.74	1	14.74	5.34	0.0541	-
X_2_^2^	116.31	1	116.31	42.13	0.0003	-
Residual	19.33	7	2.76	-	-	-
Lack of fit	12.11	3	4.04	2.24	0.2260	Not significant
Pure error	7.21	4	1.80	-	-	-
Cor total	256.70	12	-	-	-	-

**Table 7 materials-15-06997-t007:** Characteristic absorption peaks of the FTIR spectra of PBG, paraffin, ESPBG, and ESPBG after thermal cycling.

Material	Characteristic Peak Values (cm^−1^)	Sharp Peaks (cm^−1^)	Smooth Peaks (cm^−1^)
Paraffin	2953.55, 2911.87, 1458.11, 1326.50	2953.55, 2911.87	1458.11, 1326.50
PBG	3546.27, 3403.62, 1684.33, 1618.23, 1127.10, 659.24, 601.33	1684.33, 1618.23, 659.24, 601.33	3546.27, 3403.62, 1127.10
ESPBG	3547.84, 3408.20, 2920.13, 2849.80, 1685.94, 1621.58, 1459.22, 1328.62, 1141.76, 670.91, 602.50	2920.13, 2849.80, 1685.94, 1621.58, 670.91, 602.50	3547.84, 3408.20, 1459.22, 1328.62, 1141.76
ESPBG after thermal cycling	3547.84, 3409.10, 2920.13, 2849.80, 1684.25, 1612.47, 1460.20, 1327.42, 1139.26, 669.27, 602.50	2920.13, 2849.80, 1684.25, 1612.47, 1460.20, 669.27, 602.50	3547.84, 3409.10, 1327.42, 1139.26

## Data Availability

Not applicable.

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
