# Peer review of "Preparation and Pore Structure of Energy-Storage Phosphorus Building Gypsum"

_materials, 2022, doi:10.3390/ma15196997_

Round 1

Reviewer 1 Report

The present manuscript increases the number of papers related to the use of petroleum-based PCMs. I suggest providing more information that may help to stress either the novelty or contribution of the paper,  for instance :

a ) what is the chemical composition of the Air-entraining agent since it is strange that in Figure 3. XRD patterns, mainly for the porous phosphorus-building gypsum (PPBG) and the ESPBG , no chemical copounds are related to this.  

b. Section 3.9 is named: FT-IR Analysis of PBG, PPBG and ESPBG however no coments at all are given to mark diffrences or similarities between the FT-IR spectra of PPBG and PBG , ESPBG.

c. It is suggested to increase the number of references since knowledge is missed, for instance 

  • Thermal Energy Storage by the Encapsulation of Phase Change Materials in Building Elements, Materials 2021, 14
  • Development and characterization of an inorganic foam obtained

by using sodium bicarbonate as a gas generator, Construction and Building Materials 19 (2005) 543–549

Reviewer 2 Report

The proposed work focuses on the preparation and pore structure of energy storage phosphorus building gypsum. It is of potential interest to Materials journal readers.

Despite the importance of the subject addressed, this work needs many improvements to be ready for the publication in the Materials journal. 

Specific points of improvement :

-       -  Abstract is too long. So, it must be rewritten

-        - Literature review section must be improved by more previous researches.

-        -The objective of this research must be more developed.

-      -  How you can explain that the XRD analysis showed that the ESPBG was only a combination of paraffin and PPBG and no new substances were produced?

-       - Explain why the heat energy values of the ESPBG in the endothermic and freezing stages are equal0..

-        -Quality of figures must be improved.

-        -All test standards must be indicated in the manuscript.

-        -The conclusion section is too long. Only the main results must be indicated.

Reviewer 3 Report

The study is good and results are well presented. However, are there any study limitations? If yes, they have not been reported.

Reviewer 4 Report

In this paper, the pore structure of phosphorous gypsum hardening body is improved by optimizing air entraining agent and water-paste ratio. The compressive strength, microstructure and chemical properties were measured by various test methods. With the increasing demand for energy storage materials, the purpose and meaning of this study are practical. It has good research value and development prospects. The development law of the properties of phosphor building gypsum hardening body is obtained, but the analysis part can still be further improved. My personal suggestion is to further combine the microscopic results with the macroscopic properties. All in all, the research object has practical value, and the test method is novel and comprehensive. This paper needs to improve the relevant supporting content in the research motivation part. On the whole, I tend to accept it.

The following suggestions are shared for authors:

INTRODUCTION:

The introduction introduces the development and application of the phosphorous building gypsum hardened body. However, the researches on microstructure is insufficient, which makes the research motivation slightly insufficient The applicability and shortcomings of using response surface method to simplify the experimental process should be explained more comprehensively. Therefore, the introduction part is supplemented with corresponding supporting materials.

EXPERIMENTAL PROGRAM:

- In the part of materials and methods, the basis of the method used to prepare phosphorus building gypsum slurry shall be stated.

-The test steps are too simple and should be described in detail.

RESULTS AND DISCUSSION:

- The Results and discussion section well describes the test results of various properties of materials. However more words should be used to introduce the relationship between macroscopic and microscopic properties.

Reviewer 5 Report

This work is about the physical, mechanical and thermal properties of phosphorous gypsum with paraffine as energy storage material. In my opinion, the article can be published in Materials and only minor comments can be attached

- In Abstract Section: some comments about the single-factor experiment must be included so that readers can get a better idea of ​​what the article is about

- In Introduction Section: some comments about the phosphorus gypsum can be included (production, why is useful instead of white gypsum, previous studies about its use…..). Lot of readers could confuse phosphorus gypsum with phospho-gypsum, which present many environmental problems (is a simulation?), as the authors describe the production of phosphorus gypsum in article, this problems cannot be observed.

- In Materials and Methods section:

or Line 117: company of air-entraining agent?

- in table 2 the mixtures without paraffin are shown, but which is the dosage of paraffin is added in ESPGB?

-In pore size distribution analysis: a relation between mechanical and pore structure should be included.

For future works, the range of pores using a Mercury Intrusion Porosimetry (MIP) should be carried out, in range of sizes than MIP show, very interesting results could be observedThis work is about the physical, mechanical and thermal properties of phosphorous gypsum with paraffine as energy storage material. In my opinion, the article can be published in Materials and only minor comments can be attached

- In Abstract Section: some comments about the single-factor experiment must be included so that readers can get a better idea of ​​what the article is about

- In Introduction Section: some comments about the phosphorus gypsum can be included (production, why is useful instead of white gypsum, previous studies about its use…..). Lot of readers could confuse phosphorus gypsum with phospho-gypsum, which present many environmental problems (is a simulation?), as the authors describe the production of phosphorus gypsum in article, this problems cannot be observed.

- In Materials and Methods section:

or Line 117: company of air-entraining agent?

- in table 2 the mixtures without paraffin are shown, but which is the dosage of paraffin is added in ESPGB?

-In pore size distribution analysis: a relation between mechanical and pore structure should be included.

For future works, the range of pores using a Mercury Intrusion Porosimetry (MIP) should be carried out, in range of sizes than MIP show, very interesting results could be observedThis work is about the physical, mechanical and thermal properties of phosphorous gypsum with paraffine as energy storage material. In my opinion, the article can be published in Materials and only minor comments can be attached

- In Abstract Section: some comments about the single-factor experiment must be included so that readers can get a better idea of ​​what the article is about

- In Introduction Section: some comments about the phosphorus gypsum can be included (production, why is useful instead of white gypsum, previous studies about its use…..). Lot of readers could confuse phosphorus gypsum with phospho-gypsum, which present many environmental problems (is a simulation?), as the authors describe the production of phosphorus gypsum in article, this problems cannot be observed.

- In Materials and Methods section:

or Line 117: company of air-entraining agent?

- in table 2 the mixtures without paraffin are shown, but which is the dosage of paraffin is added in ESPGB?

-In pore size distribution analysis: a relation between mechanical and pore structure should be included.

For future works, the range of pores using a Mercury Intrusion Porosimetry (MIP) should be carried out, in range of sizes than MIP show, very interesting results could be observed

Round 2

Reviewer 1 Report

The comments and suggestions were partially attended. Therefore, it is petty since the missing discussion and information could benefit readers in PCMs. So, this manuscript increases the number of papers on petroleum-based PCMs with no relevant contribution

Reviewer 2 Report

The proposed work focuses on the preparation and pore structure of energy storage phosphorus building gypsum. It is of potential interest to Materials journal readers.

Despite the importance of the subject addressed, this work needs many improvements to be ready for the publication in the Materials journal. 

I think that the revised version of the submitted paper is well improved by considering the reviewers and editor recommendations and remarks. Indeed, I think that this paper is accepted in this form.
